# LEARNING TO SELECT IN-CONTEXT EXAMPLES FROM REWARD

## ABSTRACT

Large language models (LLMs) have impressive in-context learning ability. When prompted with a few examples of the same task, LLMs can solve new questions without task-specific training, demonstrating their ability of in-context learning. Recent studies revealed that the selection of contexts can significantly affect the LM's answer quality. In this work, we propose Reward-Guided Example Selection(ReGES), a novel method that learns to iteratively select in-context examples conditioned on the input question from feedback. Given a task and an example set, we use the MCTS algorithm to select different in-context examples, collect the LLM's outputs, and evaluate their accuracies. Then, we leverage the offline RL algorithm to train a value function to estimate the reward from in-context learning. During inference, we iteratively select a sequence of in-context examples for the given question based on the prediction of the value function. Our method substantially improves the performance of several LLMs (Vicuna, LLaMA-2, GPT3.5) on four benchmarks (GSM8K, Strategy QA, TREC, QNLI), and can be combined with in-context example retrieval method to give further improvement. When combined with BM25, ReGES achieves up to $+6.6$ accuracy improvement with an average of $+2.25$ over strong baselines. Moreover, we observe consistent improvement while applying the in-context examples selected by our method to language models that are not used during the training phase, demonstrating its generalization ability.

## 1 INTRODUCTION

Recent Transformer-based (Vaswani et al., 2017) large language models (LLMs) show impressive ability in various language tasks. However, further improving off-the-shelf LLMs' performance on specific tasks is still required in many practical scenarios. One solution is to fine-tune LLMs with more training data, which could substantially improve the model's performance. However, this approach is computationally expensive, sometimes even infeasible, since some models are not available for fine-tuning. Gathering enough annotations for fine-tuning is also expensive and sometimes infeasible for low-resource tasks.

Another way to improve the model's performance is in-context learning (ICL) (Wang et al., 2023; Rubin et al., 2021; Fu et al., 2022). ICL leverages LLMs' ability to learn from only a few examples in the prompt to solve new problems of the same task without additional training. This method enables fast and cheap adaptation of LLMs to new tasks. However, the number of examples that can be put into the prompt is limited by the context lengths of the LLMs, and the performance of ICL is usually inferior to fine-tuning approaches. Some recent works (Rubin et al., 2021; Liu et al., 2021; Wu et al., 2022) have shown that different choices of in-context examples can significantly affect the output quality. When given good and carefully chosen contexts, an LLM's performance can match the performance of an LLM that is fine-tuned with more data. On the other hand, when given bad contexts, an LLM's output may be almost random (Zhao et al., 2021; Lu et al., 2021; Gao et al., 2020). Therefore, it is crucial to select useful examples to achieve a good ICL performance.

Existing work has employed a wide range of methods to select good in-context samples for ICL, including searching, heuristic-based (Fu et al., 2022; Wu et al., 2022), retrieval (Rubin et al., 2021; Wang et al., 2023), and so on. Specifically, Ye et al. (2023) model the correlations between the selected examples, but the potentially complex semantic relationship between examples remains un-

explored. Zhang et al. (2022) use offline reinforcement learning to train models that select examples iteratively. However, they use a simple MLP for value predictions. As a result, their state representation contains no semantic information about the selected examples, which restricts their approach to classification tasks and smaller language models.

In this paper, we propose **Re**ward-**G**uided **E**xample **S**election (ReGES), an algorithm that iteratively adds useful examples to the context to achieve high-performance in-context learning. Specifically, we formulate the context selection problem as a sequential decision problem, where a state is the current context, which contains zero or more selected examples; an action is adding an example to the context. To solve this problem, we follow the following three steps. 1) We first generate sequences of in-context examples for each question, get the LLM's answers to the question conditioned on the examples, and evaluate the quality of the answers. The samples are collected using the Monte-Carlo tree search (MCTS) algorithm to ensure that they contain enough high-quality example sequences that are useful for training in the following step. 2) We then leverage an offline RL method (Sutton & Barto, 2018; Levine et al., 2020) and a contrastive loss (Chen et al., 2020) to learn a value function to estimate

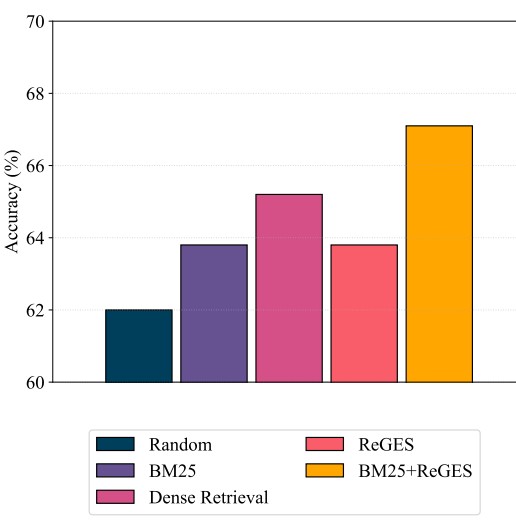

Figure 1: LLaMA2 13B performance significantly improves after applying our method.

the expected success rate for a given question and a sequence of examples. 3) Finally, during inference, given a new question, we iteratively call the value function to predict the next most useful example and add it to the context, until the model selects a terminal action or reaches the maximum number of examples.

To validate the effectiveness of our method, we empirically evaluate our method on four datasets (GSM8K, Strategy_QA, TREC, and QNLI) using three LLM families (Vicuna, LLaMA-2, and GPT). Results show that our method consistently improves the LLMs' performance on various NLP tasks. Furthermore, our method can be further improved when combined with other retrieval-based methods (i.e., BM25) to filter a subset of candidates from a large candidate pool and rerank with our value function.

Our contributions are summarized as follows:

- By formulating the example selection problem for in-context learning as a sequential decision-making problem, we propose a new method, Reward-Guided Example Selection (ReGES), that iteratively selects examples to maximize the reward of answering the question;

- We propose an MCTS-based sampling method to collect the LLM's answer quality given different in-context examples, and subsequently use an offline RL method to train the value function to estimate the expected reward for a question and a set of in-context examples;

- We show that our method consistently improves LLM's in-context learning performance on different NLP tasks, from comparatively easy natural language inference to complicated multi-step reasoning, and is generalizable to other LLMs.

## 2 RELATED WORK

**In-Context Learning** Large language models are known to have impressive in-context learning abilities. Recent LLMs, like GPT series (Brown et al., 2020; OpenAI, 2023) and the LLaMA family (Touvron et al., 2023a; Chiang et al., 2023), are able to solve new questions without task-specific training when only prompted with a few examples of the same task, which has motivated the research community to explore the area of in-context learning. One possible direction is to interpret

the mechanics behind such ability, where Xie et al. (2021) consider in-context learning as implicit Bayesian inference, while Dai et al. (2022) explain language models as meta-optimizers and understand in-context learning as implicit finetuning. Another direction is to improve in-context ability, which our work falls into by selecting a better set of in-context examples.

**Example Selection**   There are some existing methods for selecting in-context examples. Fu et al. (2022) design a heuristic criterion for tasks with chain-of-thought answers, by selecting the most complex samples (the ones with the most reasoning steps) as context, and empirically confirm that this simple heuristic can improve the quality of outputs in several reasoning datasets. Ye et al. (2023) propose compositional exemplars for in-context learning, which leverage the determinantal point process (DPP) algorithm to select a diverse yet relevant set of examples using a contrastive loss. Wu et al. (2022) propose a select-then-rank framework, where in the reranking phase, they prioritize the examples that make the LLM more confident. Rubin et al. (2021) and Wang et al. (2023) uses different methods to train a dense retriever, and use it to retrieve examples. Zhang et al. (2022) use offline reinforcement learning to train an MLP scoring function for examples, and iteratively select the example with the highest score, but the simple design of its model and state restricts its effectiveness within smaller LMs, and the performance improvement diminishes in GPT3. Wang et al. (2023) also follows an iterative process for example selection. However, the main difference is that they iteratively train a dense retriever that selects useful examples. In each iteration, the retriever retrieves examples for the questions in the training set, uses them as input to the LLM, and evaluates the outputs' quality by their log-likelihood of generating the ground truth. This additional data is used to further train the retriever for the next iteration.

## 3   PRELIMINARY

### 3.1   BACKGROUND

**In-Context learning**   In-context learning is a learning approach that enables a model to learn from the input, without fine-tuning the model. For a given target question $q$ that we want the LLM to answer, we provide a context $c$ that contains helpful information to answer $q$. We use $E + q$ as the input to the LLM, where $+$ denotes string concatenation. The context $E$ is comprised of zero or more examples, $E = e_0 + e_1 + \ldots$, where $e_i$ denotes an example question-answer pair. The examples are selected from an example pool $\mathcal{E}$ available to the LLM, that is, $e_i \in \mathcal{E}$ for all $e_i$ in $E$.

**Reinforcement learning**   In this paper, we consider a sequential decision-making approach to example selection. We formulate the example selection problem as a Markov decision process. Concretely, a state is the target question that the LLM needs to answer and a set of (zero or more) examples that we already selected from the example pool; an action is the next example to select. The reward is defined by the quality of the LLM output using the selected examples as the context. The goal of this sequential decision-making problem is to select examples incrementally so that when the selected examples are used as context, the LLM generates a high-quality response.

### 3.2   PROBLEM FORMULATION

We are now ready to formally define the example selection problem in in-context learning that we address in this paper. Given the target question $q$, we want to optimally select examples from an example set, and use them as context in the input to the LLM in order to optimize the quality of the output. Due to the context length of the LLM, we consider selecting up to $N$ examples as the context, where $N$ is a pre-defined parameter. The output of the LLM is evaluated by a task-dependent score function, denoted by $\mathbf{S}$. The score function is the LLM's answer accuracy in most cases. The objective of the example selection problem is to find the best composition of examples from the pool to maximize the score, that is,

$$\arg\max_{e_0, e_1, \ldots, e_N \in \mathcal{E}} \mathbf{S}(LLM(e_0 + e_1 + \cdots + e_N + q)),$$

where $LLM(e_0 + e_1 + \cdots + e_N + q)$ denotes the LLM's output given the corresponding input. In this paper, we focus on a sequential decision-making approach that selects $e_0, e_1, \ldots, e_N$ sequentially.

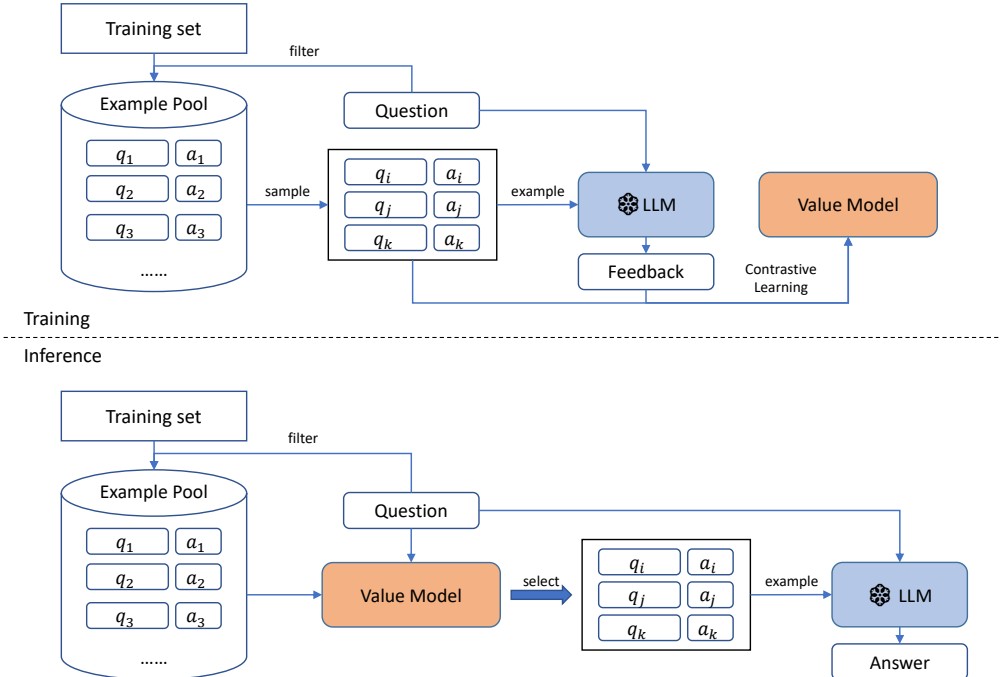

Figure 2: The overall pipeline of ReGES. (Top) Training: We use MCTS to collect examples, use them as context, and obtain feedback on the LLM's output quality. We use the context-feedback pairs to train the value model. (Bottom) Inference: Given a target question that the model needs to answer, our framework uses the value model to iteratively select question-answer pairs from the example pool and add them to the context.

# 4 REWARD-GUIDED EXAMPLE SELECTION

In this section, we describe our Reward-Guided Example Selection algorithm (ReGES) that solves the context selection problem in in-context learning. The overall framework is shown in Figure 2.

## 4.1 OVERVIEW

As in-context learning's performance can be affected by the complex semantic relations between the examples and the question, we use another language model to capture the relations. Specifically, we use a transformer encoder with an MLP head as a value model to predict the quality of the output of an LLM using a sequence of examples as the context. We initialize our value model from FLAN-T5 large encoder since it uses relative positional embedding that supports a larger input length. For each text input $I = e_0 + e_1 + \cdots + e_M + q$, the value model gives a score $S = V(I)$, which is the estimated quality of the model's output given in-context examples $e_0 + e_1 + \cdots + e_M$ and the question $q$.

## 4.2 TRAINING

We train the model in an offline RL pipeline. We first generate the training data by collecting trajectories $\{I_i\}$ and corresponding rewards $\{r_i\}$. Here, the trajectories are lists of selected examples used as the context, and the rewards are the accuracies of the LLM's outputs. The lengths of the trajectories range from 1 to the maximum of the examples allowed, $N$. We could certainly generate the sequences randomly. However, random examples are unlikely to help the LLM generate correct solutions, which makes most of the training data useless. We instead use the Monta-Carlo tree search (MCTS) algorithm to generate the trajectories, similar to Guo et al. (2014). In this way, more

trajectories with higher rewards will be generated, and the training data will be more balanced in terms of their rewards. Additionally, to make sure that trajectories of different lengths are generated, the termination action is always considered by the tree search algorithm. This helps the value model understand the effects of using different numbers of examples.

The MCTS algorithm keeps the average return starting from the $I_i$ by either continuing to select examples or terminating immediately. We denote the return by $R_i$. We want the value function to estimate $R_i$ accurately. To train the value model, we use a combination of two loss functions. First, to estimate the return accurately, we employ a binary cross-entropy (BCE) loss function as follows,

$$L_{BCE} = -\big[R_i \log\big(\sigma(V(I_i))\big) + (1 - R_i) \log\big(1 - \sigma(V(I_i))\big)\big].$$

Second, when we use the value model, it is crucial that the model makes the right decision on predicting the next useful example correctly given a prefix of examples. In light of this observation, we use an InfoNCE-based contrastive loss (Chen et al., 2020) to make sure the value function distinguishes good examples from bad ones. Specifically, we sort all the trajectories in lexicographical order so that adjacent trajectories share a common prefix. We then split all the trajectories into batches. Within a batch, we denote the trajectories and their rewards by $(I, r)$. We regard the $k$ examples with the highest returns in this batch as positive examples $\{(I_i^+, r_i^+)\}_{i=1}^k$, and others as negative examples. Here, $k$ is a predefined hyperparameter. We train the value function using the following contrastive loss:

$$L_{cont} = -\log \frac{\sum_{i=1}^k e^{V(I_i^+)}}{\sum_{i=1}^N e^{V(I_i)}}.$$

Finally, the loss function $L$ we use to train the value model is a weighted sum of the BCE loss and the contrastive loss:

$$L = \alpha L_{cont} + L_{BCE}.$$

### 4.3 INFERENCE

Given a question $q$ and a LLM, we iteratively call our value model $V$ to get the final selection. At each step during selection, suppose we have already selected $k$ examples $e_0, e_1, \cdots, e_k$ in the previous iterations, and candidate examples for this iteration are $c_1 \cdots c_m$, the value model $V$ will take $I_i = e_0 + e_1 + \cdots + e_k + c_i + q$ as input for each candidate $c_i$, and outputs a score $S_i = V(I_i)$ on the success rate prediction of selecting $c_i$ as the next example. Then, we simply choose the example with the largest score as the next example in the context, $e_{k+1}$, and continue to the next iteration until we select the terminal action [TERM] or reach the maximum number of examples allowed. In this way, the model can have full access to the question and the examples already selected, allowing it to give scores to the candidate examples conditioning on the current information.

## 5 EXPERIMENTS

In this section, we empirically evaluate the effectiveness of ReGES with different LLMs and on various datasets, ranging from comparatively easy question-answering tasks to difficult multi-step reasoning tasks. We show that ReGES outperforms the state-of-the-art in-context learning algorithms in most settings.

### 5.1 EVALUATION SETUP

**Datasets** We consider four different datasets (GSM8K, StrategyQA, TREC, QNLI) on four different tasks. Specifically, GSM8K (Cobbe et al., 2021) is a math reasoning dataset with step-by-step answers required for chain-of-thought prompting (Wei et al., 2022). StrategyQA (Geva et al., 2021) is a commonsense reasoning dataset with supporting facts as annotations provided for each reasoning step, where we concatenate these facts with the final answer as a CoT answer. TREC (Voorhees & Tice, 2000) is a text classification dataset that classifies text questions into 6 types according to the topics. QNLI (Wang et al., 2018) is a natural language inference dataset where each example asks whether the text is the correct answer to the given question.

For GSM8K and StrategyQA, we test our method with chain-of-thought by adding "Let's think step by step." at the beginning of the output (Kojima et al., 2022). For TREC and QNLI,

since there are no chain-of-thought answers provided, we simply concatenate the question and the final answer as an example. Given that there is no official validation set for the first three datasets, we randomly select a subset of examples from the training set to form a validation split: 500 for GSM8K and TREC, and 250 for StrategyQA.

**Language Models** We tested our ReGES algorithm on three LLM families: Vicuna (Chiang et al., 2023), LLaMA-2 (Touvron et al., 2023b), and GPT (Brown et al., 2020; OpenAI, 2023), with several sizes and versions. More specifically, for Vicuna models, we use 7B and 13B on v1.1 and 33B on v1.3; for LLaMA-2, we tested over all three released sizes (7B, 13B, 70B); and for GPT models, gpt-3.5-turbo-0613 is applied. For all experiments, the value model is initialized from the Flan-T5 large's encoder (Chung et al., 2022), an instruction fine-tuned encoder with 340M parameters.

**Baselines** We compare ReGES with three baseline algorithms: **Random**, **BM25**, and **Dense Retrieval**. For the Random baseline, we select examples uniformly randomly from the example pool. We also run experiments with 5 random seeds and compute the average results to reduce variance. For Dense Retrieval baseline, we use the off-the-shelf sentence transformer (all-MiniLM-L12-v2) (Reimers & Gurevych, 2019) to compute the vector representation for text, and then retrieve the closest examples. For all the baselines, the number of examples we select for each question is set to be the same as the maximum number of examples allowed for our method.

**ReGES** We tested our method under two settings, one randomly selects examples as the example pool (denoted as **ReGES**). We also consider filtering the example pool that is more relevant to the target question using BM25 (denoted as **BM25 + ReGES**). For all the datasets and the two settings, our value model is trained from the outputs of one LLM and then tested over all the other LLMs, demonstrating that the value function can be model-agnostic and used to help in-context learning on other LLMs. The LLM used to collect training data is slightly different: for GSM8K and Strategy QA, we use LLaMA-2 13B to collect the training data, since these datasets are more challenging and require chain-of-thought reasoning. For the rest of the datasets, we use Vicuna 13B. Also, the reward for the trajectories is the mean accuracy of $8$ generated answers for the GSM8K and Strategy QA since there could be multiple correct answers. For the other three datasets, we directly use the log-likelihood of generating the uniquely correct answer. More implementation details and hyperparameters are provided in Appendix A.1.

| Model | Method | GSM8K | StrategyQA |
|---|---|---|---|
| LLaMA2 7B | Random Examples | 26.6 | 65.4 |
| | BM25 | 28.4 | 66.4 |
| | Dense Retrieval | **30.2** | **68.8** |
| | ReGES | 28.4 | 68 |
| | BM25 + ReGES | 28.4 | 66.4 |
| LLaMA2 13B | Random Examples | 40.1 | 71.2 |
| | BM25 | 40.2 | 69.6 |
| | Dense Retrieval | 42.2 | 72.0 |
| | ReGES | 41.4 | 68.0 |
| | BM25 + ReGES | **43.2** | **73.6** |
| LLaMA2 70B | Random Examples | 59.0 | 74.7 |
| | BM25 | 63.4 | 78.0 |
| | Dense Retrieval | 60.4 | 78.0 |
| | ReGES | 60.8 | 78.0 |
| | BM25 + ReGES | **63.4** | **79.6** |

Table 1: Performance on LLaMA2 models For GSM8K and StrategyQA when using greedy decoding. The best result in each set of experiments is **bolded**.

## 5.2 MAIN RESULTS

Table 1, 2 show the evaluation results on the series of LLMs where we collected feedback from. We observe that BM25 serves as an overall stronger baseline than Random, while still unable to show improvements in some cases. Dense Retrieval, though retrieves examples according to representative sentence embeddings, does not show a consistent improvement over BM25. As for our method, we

can see that our ReGES consistently improves the performance compared with the corresponding baseline, sometimes by a large margin. For all reported methods, GSM8K and StrategyQA give smaller improvements than TREC and QNLI in general, potentially because these two tasks are hard reasoning tasks that rely more on LLMs' intrinsic reasoning ability and are hard to improve through in-context learning. To get examples demonstrating the hardness of GSM8K, see Appendix A.2.

| Model | Method | TREC | QNLI |
|-------|--------|------|------|
| Vicuna 7B | Random Examples | 50.6 | 59.1 |
| | BM25 | 74 | 63.2 |
| | Dense Retrieval | 71.4 | 62.6 |
| | ReGES | 63 | 61.1 |
| | BM25 + ReGES | **77.6** | **69.4** |
| Vicuna 13B | Random Examples | 65.08 | 70.28 |
| | BM25 | 81.8 | 71.4 |
| | Dense Retrieval | 79.4 | 70 |
| | ReGES | 78.8 | **75.2** |
| | BM25 + ReGES | **88.4** | 74.6 |
| Vicuna 33B | Random Examples | 72.48 | 70.62 |
| | BM25 | **87.8** | 74.2 |
| | Dense Retrieval | 84 | 72.4 |
| | ReGES | 78 | 73.8 |
| | BM25 + ReGES | 86.6 | **77.7** |

Table 2: Performance on Vicuna models For TREC and QNLI when using greedy decoding.

## 5.3 GENERALIZATION OVER LLMS

Our method is trained from feedback of only one LLM for each dataset. In Table 3, we report the performance of ReGES for all series LLMs and all datasets. Even if trained from the feedback of one LLM, then tested on other series of LLMs, ReGES still shows a general improvement with an average of +2.25 over BM25, suggesting that ReGES learns a general strategy for selecting good in-context examples, which can be transferred to other LLMs without additional training.

## 6 ANALYSIS

### 6.1 EFFECT OF THE LOSS FUNCTION

Our loss function is the combination of two separate losses: an InfoNCE loss designed for contrastive learning, and a BCE regression loss for predicting the reward. We design such a combined loss function since we need our model to identify the best examples as positive examples, while still being able to rank negative examples. We found that when only one loss is applied, the model cannot learn properly and returns outputs close to the baseline. Such ablation results, as shown in Table 4, indicate that both losses are necessary in order to learn from LLM's feedback.

### 6.2 SCALING EXAMPLE POOL SIZE

In our main results, we use an example pool size of 32 for GSM8K and StrategyQA, and 64 for TREC and QNLI. To see how the example pool size affects the performance of our method, we evaluated our method under different example pool sizes on QNLI: 16, 64 (our main result), and 256, shown in Figure 3 (left). Despite some fluctuations, a larger pool size yields better results at the cost of more computation during inference. One can choose a proper pool size to strike a balance between performance and inference cost.

### 6.3 NECESSITY OF ITERATIVE SELECTION

To validate the effectiveness of the iterative design, we also evaluated our method with the iterative part removed. More specifically, for each example, we take the average accuracy of collected trajectories containing the example as its score to the question. Then, we similarly train the value model

| Model | Method | GSM8K | StrategyQA | TREC | QNLI | Avg |
|---|---|---|---|---|---|---|
| Vicuna 7B | Random Examples | 15.8 | 63.4 | 50.6 | 59.1 | 47.2 |
| | BM25 | **19.4** | 62.4 | 74.0 | 63.2 | 54.8 |
| | Dense Retrieval | 19.2 | 62.8 | 71.4 | 62.6 | 54 |
| | ReGES | 18.2 | **63.6** | 63.0 | 61.1 | 51.5 |
| | BM25 + ReGES | 17.2 | 62 | **77.6** | **69.4** | **56.6** |
| Vicuna 13B | Random Examples | 26.88 | 64.7 | 65.1 | 70.3 | 56.7 |
| | BM25 | **30.8** | 64.8 | 81.8 | 71.4 | 62.2 |
| | Dense Retrieval | 30.4 | 64.4 | 79.4 | 70.0 | 61.2 |
| | ReGES | 30.2 | 68.8 | 78.8 | **75.2** | 63.3 |
| | BM25 + ReGES | 30.2 | **70.0** | **88.4** | 74.6 | **65.8** |
| Vicuna 33B | Random Examples | 44.96 | 71.6 | 72.5 | 70.6 | 64.9 |
| | BM25 | 48.2 | 71.2 | **87.8** | 74.2 | 70.4 |
| | Dense Retrieval | 48.0 | 71.6 | 84.0 | 72.4 | 69 |
| | ReGES | 49.0 | 69.6 | 78.0 | 73.8 | 67.6 |
| | BM25 + ReGES | **49.2** | **73.6** | 86.6 | **77.7** | **71.8** |
| LLaMA2 7B | Random Examples | 26.6 | 65.4 | 60.5 | 71.3 | 55.9 |
| | BM25 | 28.4 | 66.4 | 72.4 | 71.5 | 59.7 |
| | Dense Retrieval | **30.2** | **68.8** | 73.2 | 71.4 | 60.9 |
| | ReGES | 28.4 | 68.0 | 72.6 | **77.8** | 61.7 |
| | BM25 + ReGES | 28.4 | 66.4 | **76.8** | 77.6 | **62.3** |
| LLaMA2 13B | Random Examples | 40.1 | 71.2 | 62.9 | 73.6 | 62.0 |
| | BM25 | 40.2 | 69.6 | 76.4 | 68.9 | 63.8 |
| | Dense Retrieval | 42.2 | 72.0 | 75.6 | 70.8 | 65.2 |
| | ReGES | 41.4 | 68.0 | 72.0 | **75.1** | 64.1 |
| | BM25 + ReGES | **43.2** | **73.6** | **79.0** | 72.5 | **67.1** |
| LLaMA2 70B | Random Examples | 59.0 | 74.7 | 67.4 | 82.1 | 70.8 |
| | BM25 | 63.4 | 78.0 | 82.8 | 77.8 | 75.5 |
| | Dense Retrieval | 60.4 | 78.0 | 81.0 | 77.4 | 74.2 |
| | ReGES | 60.8 | 78.0 | 77.0 | **83.8** | 74.9 |
| | BM25 + ReGES | **63.4** | **79.6** | **85.6** | 82.3 | **77.7** |
| GPT-3.5-turbo | Random Examples | 79.8 | 72.1 | 70.4 | 79.8 | 75.5 |
| | BM25 | 80.0 | 72.4 | **83.2** | 80.0 | 78.9 |
| | Dense Retrieval | 79.0 | 73.2 | 79.6 | 80.4 | 78.1 |
| | ReGES | **81.0** | 69.2 | 71.6 | 82.6 | 76.1 |
| | BM25 + ReGES | 78.8 | **74.0** | 81.0 | **84.9** | **79.7** |

Table 3: Performance on transferred to different series of LLMs. ReGES improves over random and BM25 baselines under most cases, regardless of the evaluated LLM. The best result in each set of experiments is **bolded**.

| Model | BM25 | BCE loss | InfoNCE loss | ReGES (both) |
|---|---|---|---|---|
| Vicuna 7B | 63.2 | 60.9 | 59.7 | 69.4 |
| Vicuna 13B | 71.4 | 68.33 | 70.9 | 74.6 |
| Vicuna 33B | 74.2 | 72.7 | 69.8 | 77.7 |
| LLaMA2 7B | 71.5 | 69.9 | 70.4 | 77.6 |
| LLaMA2 13B | 68.9 | 70.4 | 67 | 72.5 |
| LLaMA2 70B | 77.8 | 78.7 | 78.7 | 82.3 |
| GPT-3.5-turbo | 80 | 78.6 | 79.2 | 84.9 |

Table 4: QNLI Validation accuracy when trained under different losses.

to learn from these scores for each example-question pair. Finally, during inference, we select those examples with the highest predicted score from the trained model as the context of the problem. As shown in Figure 3 (right), the non-iterative variation is generally inferior to ReGES, indicating that the impacts of examples on the model's output are not independent, therefore requiring considering selected examples jointly. Our iterative design exactly models such joint influences. Therefore it outperforms baseline algorithms that consider examples independently.

## 6.4 PERFORMANCE CHANGE AFTER RANDOMLY REORDERING THE EXAMPLES

To examine the effect of the order of the examples that our method selected, we additionally tested a Shuffled setting of ReGES. Under this setting, we randomly permute the examples selected by

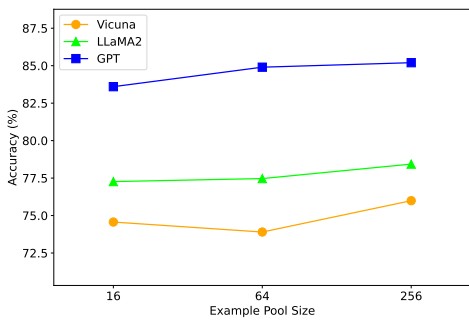 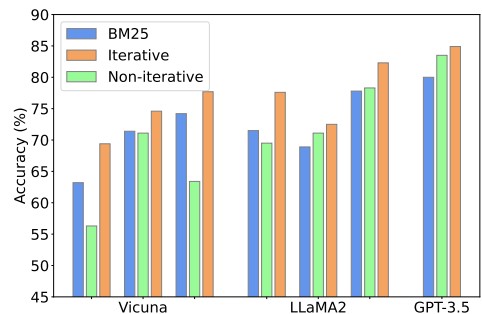

Figure 3: Left: QNLI validation accuracy under different example pool sizes. Vicuna results are the mean of 3 vicuna models, and LLaMA2 results are similarly the mean of 3 LLaMA2 models. Right: QNLI with and without iterative selection.

| Model | BM25 | BM25 + ReGES | ReGES Shuffled |
|---|---|---|---|
| Vicuna 7B | 63.2 | 69.4 | 65.8 |
| Vicuna 13B | 71.4 | 74.6 | 72.8 |
| Vicuna 33B | 74.2 | 77.7 | 73.75 |
| LLaMA2 7B | 71.5 | 77.6 | 76.5 |
| LLaMA2 13B | 68.9 | 72.5 | 74.4 |
| LLaMA2 70B | 77.8 | 82.3 | 82.4 |
| GPT-3.5-turbo | 80 | 84.9 | 83.9 |

Table 5: Accuracy after random shuffling selected examples, compared with BM25 baseline and ReGES on QNLI. Performance drops on Vicuna models, indicating that our method learns the model-specific order preferences.

ReGES before formatting these examples as context. The results are presented in Table 5. Although still better than the BM25 baseline overall, the performance improvement was reduced by a large margin for Vicuna models, while for other models the performance is still on par with our main method. This indicates that ReGES learns a model-specific good ordering, which is consistent with findings of Lu et al. (2021), that LLMs have non-transferable preferences over the order of in-context examples, allowing us to have additional advantages compared with other methods due to the order awareness nature of our iterative design.

# 7 CONCLUSION

In this paper, we introduced a neural-based method to select examples iteratively for in-context learning. This framework collects training data by calling a frozen LLM and then learns a transformer value model to give a score on current candidate examples conditioning on the target question and the selected examples in the context. During inference, we iteratively call the value model to select the next example. We conduct comprehensive evaluations of our method with multiple LLMs and on various datasets, showing that our method consistently outperforms strong baselines. Our method generalizes to other LLMs not used for collecting train data, without the need to re-train value models for different LLMs.

Currently, our method still requires a training set for collecting feedback and splitting the example pool. One possible future work is to explore the effectiveness of our method under low-resource conditions, where there are only limited labeled data and requires a generalizable model trained from other tasks.

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

# A APPENDIX

## A.1 INPLEMENTATION DETAILS

The hyperparameter used for our main results is presented in Figure 6. Due to hardware limitations, we may not collect trajectories for all training samples, but we will still start training if the collecting process is almost complete. To ensure there are enough examples in the context, we force our method to select at least 5 examples during inference, while results show that this may be suboptimal since selecting fewer examples is better in some cases.

## A.2 GSM8K FAILURE CASES

Here we provide some examples that LLaMA2 13B fails to answer correctly in GSM8K. In the first example, LLaMA2 goes wrong in arithmetic involving fractions. In the second and third examples, LLaMA2 fails to figure out the correct process of solving the problem. We can see that, correctly answering GSM8K questions requires strong arithmetic ability and mathematical reasoning ability, which are hard to improve through in-context learning.

|                                      | GSM8K         | StrategyQA    | TREC          | QNLI          |
|--------------------------------------|---------------|---------------|---------------|---------------|
| max #examples selected               | 5             | 6             | 8             | 8             |
| example pool size                    | 32            | 32            | 64            | 64            |
| model used to collect feedback       | LLaMA2 13B    | LLaMA2 13B    | Vicuna 13B    | Vicuna 13B    |
| #trajectories sampled                | 128           | 128           | 160           | 128           |
| #answers generated for each trajectory | 8           | 8             |               |               |
| temperature used to collect feedback | 0.5           | 0.5           |               |               |
| MCTS exploration constant            | $0.25\sqrt{2}$ | $0.25\sqrt{2}$ | $0.25\sqrt{2}$ | $0.25\sqrt{2}$ |
| value model MLP layer size           | 1024          | 1024          | 1024          | 1024          |
| k in contrastive loss                | 4             | 4             | 4             | 4             |
| batch size                           | 64            | 64            | 64            | 64            |
| max train steps                      | 18000         | 18000         | 18000         | 18000         |
| weight decay                         | 0.004         | 0.004         | 0.004         | 0.004         |

Table 6: Detailed hyperparameters used in our experiment.

| Question | Generated Answer |
|----------|------------------|
| TreQuan is throwing rocks in the river and he notices that the bigger the rock, the wider the splash. Pebbles make a splash that is a 1/4 meter wide. Rocks make a splash that is 1/2 a meter wide, and boulders create a splash that is 2 meters wide. If he tosses 6 pebbles, 3 rocks, and 2 boulders, what is the total width of the splashes he makes? | The pebbles make a splash that is 6 * 1/4 = 1 1/2 meters wide. The rocks make a splash that is 3 * 1/2 = 3 meters wide. The boulders make a splash that is 2 * 2 = 4 meters wide. The total width of the splashes is 1 1/2 + 3 + 4 = 9 meters. The answer is 9 |
| Julie, Letitia and Anton decided to have dinner together at a nearby restaurant. Julie orders $10 worth of food, Letitia orders $20 worth of food, and Anton orders $30 worth of food. How much should each pay in tips if they want to split a 20% tip for the whole meal evenly? | First, we find the total cost of the meal by adding the amounts that each person ordered: 10+20+30=60. Then we divide this total by 3, as there are three people eating, finding a per-person cost of 60/3=20. Then we find the total tip amount for a 20% tip, by multiplying the per-person cost by 20/100, finding 20*20/100=4 per person. Finally, we divide the total tip amount by 3, as they want to split it evenly, finding a per-person tip of 4/3=1.33. So each person should pay 1.33 in tips. The answer is 1.33 |
| Frankie's parents let him have many pets. He has six more snakes than he has cats. He has one less parrot than cats. Six of his pets have four legs. He has 2 dogs. How many pets does he have in total? | Let x be the number of cats he has. He has x + 6 snakes. He has x - 1 parrot. The total number of pets is x + 6 + x - 1 = 2x + 5. He has 2 dogs. The total number of pets is 2x + 5 + 2 = 2(x+5) + 2. The answer is 2x + 7 |

Table 7: LLaMA2 13B failure cases on GSM8K.

