# OpenReview forum: "Learning to Select In-context Examples from Reward"
_ICLR.cc/2024/Conference — ICLR 2024 Conference Withdrawn Submission_

### Official Review · Reviewer_GVqZ · 2023-10-27

**Soundness:** 3 good
**Presentation:** 3 good
**Contribution:** 2 fair
**Rating:** 5
**Confidence:** 4

**Summary:**

The paper propose a method ReGES to better organize in-context exemplars through MCTS searching. The searching is guided by a value model that is trained with contrastive learning.
To show the efficacy of ReGES,  ReGES is evaluated on various LLM (i.e., Vicuna, LLaMA-2, GPT3.5) and datasets (i.e., GSM8K, Strategy QA, TREC, QNLI).

**Strengths:**

- Using MCTS to search in-context examples is an intuitive and effective idea.
- The paper is mostly well written, with intensive experiments and analysis on various aspects of the proposed method.

**Weaknesses:**

- The author mentioned the generalization ability across models, where ReGES trained on data collected from LLaMA-13B can be applied to other models, but it's unclear whether ReGES can be generalized across datasets. This generalization ability is not trivial as many benchmarks (e.g., BIG-bench, MMLU) consists of dozens of tasks and it would be time-consuming to train a value model for each of these tasks.
- It's better to consider the inference latency when applying MCTS compared to other baselines.
- It's also unknown how the performance will change if we replace the value model from Flan-T5 with another.

**Questions:**

- How about the generalization ability across datasets?
- How about the inference latency compared with baselines?
- Will the results vary much if we choose another backbone for value model?

Minor suggestion: It's better if the format (e.g., bolded or not, decimal precision) of results in and across tables is consistent.

**Details Of Ethics Concerns:**

None.

---

### Official Review · Reviewer_PmkA · 2023-10-29

**Soundness:** 2 fair
**Presentation:** 2 fair
**Contribution:** 2 fair
**Rating:** 5
**Confidence:** 4

**Summary:**

This work focuses on the challenge of improving the in-context learning ability of LLMs by proposing a novel method called Reward-Guided Example Selection (ReGES). The main contribution of this work is the development of an iterative example selection process that leverages the MCTS algorithm and offline RL to train a value function for estimating rewards from in-context learning. The proposed approach demonstratesperformance improvements on several LLMs and benchmarks, and exhibits generalization ability.

**Strengths:**

1. This work demonstrates the effectiveness of ReGES by applying it to several LLMs, including Vicuna, LLaMA-2, and GPT3.5, and evaluating its performance on four benchmarks: GSM8K, Strategy QA, TREC, and QNLI. The results show improvements in performance compared to baseline methods.

2. This work reveals the generalizablity of ReGES. The authors observe consistent improvement when applying the in-context examples selected by ReGES to language models that were not used during the training phase. This finding suggests that the proposed method have potential to be effectively applied to a wide range of LLMs and tasks.

**Weaknesses:**

1. Lack of comparison with stronger baselines such as [1, 2, 3, 4]. Especially, [1] also applies an RL-based method to select in-context examples. At least you should conceptually compare with these existing work.

[1] Y. Zhang, Shi Feng, and Chenhao Tan. "Active example selection for in-context learning."

[2] Gabriel Poesia, et al. "Synchromesh: Reliable code generation from pre-trained language models."

[3] S. An, et al. "Skill-Based Few-Shot Selection for In-Context Learning."

[4] X. Li and X. Qiu. "Finding Support Examples for In-Context Learning."

**Questions:**

1. Comparing with other training-based selection methods such as EPR[5], TST[2] and CEIL[6], what is the advantage of ReGES?

[5] Ohad Rubin, Jonathan Herzig, and Jonathan Berant. "Learning to retrieve prompts for in-context learning."

[6] J. Ye, et al. "Compositional exemplars for in-context learning."

---

### Official Review · Reviewer_vcLv · 2023-10-29

**Soundness:** 2 fair
**Presentation:** 2 fair
**Contribution:** 2 fair
**Rating:** 3
**Confidence:** 4

**Summary:**

This paper proposes a new method to iteratively select in-context learning examples for a given task. The method consists of two steps: (1) using the MCT5 algorithm to generate a set of examples-performance pairs, and (2) using the generated data to train a value function that can estimate the reward of using different in-context examples for a given input. At inference time, the method selects the optimal in-context learning examples based on the value function. The authors conduct experiments on four datasets (GSM8K, Strategy QA, TREC, and QNLI) with several models (Vicuna, LLaMA-2, and GPT3.5).

**Strengths:**

1. This paper is well-structured and easy to understand.
2. The main novelty of this paper is to formulate the problem of selecting in-context learning examples as a sequential decision-making problem. This is a reasonable approach, considering that the order of examples also affects the ICL performance [1].
3. The experimental results on different datasets and models demonstrate that the proposed iterative selection method can improve the performance in some cases/scenarios.

[1] Lu, Y., Bartolo, M., Moore, A., Riedel, S., & Stenetorp, P. (2021). Fantastically ordered prompts and where to find them: Overcoming few-shot prompt order sensitivity. arXiv preprint arXiv:2104.08786.

**Weaknesses:**

1. My main concern about this paper is the application range of the proposed method. As far as I understand, the method requires training a separate “value function” for each dataset, which involves both data generation and model training. Moreover, when using the MCTS algorithm to generate the examples-performance pairs, it has to make inference for each combination. This seems to be costly in terms of both tokens and GPU fees, especially in the time when using LLMs as a service.
2. When making predictions, the proposed method needs to iteratively select question-answer pairs from the example pool using the “value function”. This introduces a latency in inference time. The authors should provide some information about the cost of their method.
3. The experimental evaluation is insufficient. The authors only perform experiments on four datasets, which is not enough to draw a solid conclusion. More experiments should be conducted and compared with related work [2][3].
4. The experiments should also include out-of-distribution tasks, especially considering that the “value function” needs to be trained for every dataset. The authors at least needs to justify when OOD happens, the “value function” still works.

[2] Li, X., & Qiu, X. (2023). Finding supporting examples for in-context learning. arXiv preprint arXiv:2302.13539.

[3] Chang, T. Y., & Jia, R. (2023, July). Data curation alone can stabilize in-context learning. In Proceedings of the 61st Annual Meeting of the Association for Computational Linguistics (Volume 1: Long Papers) (pp. 8123-8144)

**Questions:**

Can you provide the information for both the "value function" training cost and inference latent at test time?

---

### Official Review · Reviewer_dZ1D · 2023-11-05

**Soundness:** 2 fair
**Presentation:** 3 good
**Contribution:** 2 fair
**Rating:** 5
**Confidence:** 3

**Summary:**

The paper proposes Reward-Guided Example Selection (ReGES), a novel method that learns to iteratively select in-context examples conditioned on the input question from feedback. Given a task and an example set, the paper proposes to use the MCTS algorithm to select different in-context examples, collect the LLM’s outputs, and evaluate their accuracies. Then, the offline RL algorithm is leveraged to train a value function to estimate the reward from in-context learning. During inference, a sequence of in-context examples are iteratively selected for the given question based on the prediction of the value function.

**Strengths:**

(1) The research topic is an emerging research topic.

(2) The developed algorithm has good intuitions.

(3) The paper is very well written and presented.

(4) The proposed method is evaluated on multiple LLMs, demonstrating its generalization ability.

**Weaknesses:**

(1) The compared baselines may be simple. Actually, these baselines seems to be general, which are for general retrieval tasks instead of specially tailored to prompt selection task.

(2) The studied topic is very popular with many previous works. This paper seems to only cover some of the previous works. It will be great to compare or discuss these previous works, so the position of this paper is more clear.

Hongjin, S. U., et al. "Selective Annotation Makes Language Models Better Few-Shot Learners." The Eleventh International Conference on Learning Representations. 2022.

Lee, Young-Jun, Chae-Gyun Lim, and Ho-Jin Choi. "Does GPT-3 generate empathetic dialogues? A novel in-context example selection method and automatic evaluation metric for empathetic dialogue generation." Proceedings of the 29th International Conference on Computational Linguistics. 2022.

Gupta, Shivanshu, Sameer Singh, and Matt Gardner. "Coverage-based Example Selection for In-Context Learning." arXiv preprint arXiv:2305.14907 (2023).

Yang, Sohee, et al. "Improving Probability-based Prompt Selection Through Unified Evaluation and Analysis." arXiv e-prints (2023): arXiv-2305.

An, Shengnan, et al. "Skill-Based Few-Shot Selection for In-Context Learning." arXiv preprint arXiv:2305.14210 (2023).

Kumar, Aswanth, et al. "In-context Example Selection for Machine Translation Using Multiple Features." arXiv preprint arXiv:2305.14105 (2023).

Shen, Lingfeng, et al. "Flatness-Aware Prompt Selection Improves Accuracy and Sample Efficiency." arXiv preprint arXiv:2305.10713 (2023).

Shum, KaShun, Shizhe Diao, and Tong Zhang. "Automatic Prompt Augmentation and Selection with Chain-of-Thought from Labeled Data." arXiv preprint arXiv:2302.12822 (2023).

Agrawal, Sweta, et al. "In-context examples selection for machine translation." arXiv preprint arXiv:2212.02437 (2022).

Liao, Chonghua, Yanan Zheng, and Zhilin Yang. "Zero-Label Prompt Selection." arXiv preprint arXiv:2211.04668 (2022).

The idea of improving prompts with reinforcement learning also exists in previous papers, for example,

Deng, Mingkai, et al. "RLPrompt: Optimizing Discrete Text Prompts with Reinforcement Learning." Proceedings of the 2022 Conference on Empirical Methods in Natural Language Processing. 2022.

**Questions:**

(1) It seems that all the baselines are for general retrieval purpose, and none of them are specially tailored to prompt selection task? If there any reason why these baselines (which are specially tailored to prompt selection task) are not included in the comparison? Even if the settings are not exactly the same, might some baselines be adjusted to be applied in the setting studied in this paper? If the paper studies a unique novel setting where no previous papers' approaches can be used, can you previous justify why the novel setting is practical and well-motivated (by some real-world use case or case study)?

(2) Can you please describe the differences between this paper and the following papers? Even some papers may not directly compared in the experiments, some discussions will be greatly appreciated.

Hongjin, S. U., et al. "Selective Annotation Makes Language Models Better Few-Shot Learners." The Eleventh International Conference on Learning Representations. 2022.

Lee, Young-Jun, Chae-Gyun Lim, and Ho-Jin Choi. "Does GPT-3 generate empathetic dialogues? A novel in-context example selection method and automatic evaluation metric for empathetic dialogue generation." Proceedings of the 29th International Conference on Computational Linguistics. 2022.

Gupta, Shivanshu, Sameer Singh, and Matt Gardner. "Coverage-based Example Selection for In-Context Learning." arXiv preprint arXiv:2305.14907 (2023).

Yang, Sohee, et al. "Improving Probability-based Prompt Selection Through Unified Evaluation and Analysis." arXiv e-prints (2023): arXiv-2305.

An, Shengnan, et al. "Skill-Based Few-Shot Selection for In-Context Learning." arXiv preprint arXiv:2305.14210 (2023).

Kumar, Aswanth, et al. "In-context Example Selection for Machine Translation Using Multiple Features." arXiv preprint arXiv:2305.14105 (2023).

Shen, Lingfeng, et al. "Flatness-Aware Prompt Selection Improves Accuracy and Sample Efficiency." arXiv preprint arXiv:2305.10713 (2023).

Shum, KaShun, Shizhe Diao, and Tong Zhang. "Automatic Prompt Augmentation and Selection with Chain-of-Thought from Labeled Data." arXiv preprint arXiv:2302.12822 (2023).

Agrawal, Sweta, et al. "In-context examples selection for machine translation." arXiv preprint arXiv:2212.02437 (2022).

Liao, Chonghua, Yanan Zheng, and Zhilin Yang. "Zero-Label Prompt Selection." arXiv preprint arXiv:2211.04668 (2022).

Deng, Mingkai, et al. "RLPrompt: Optimizing Discrete Text Prompts with Reinforcement Learning." Proceedings of the 2022 Conference on Empirical Methods in Natural Language Processing. 2022.